# Prevalence of Diarrhoeagenic *Escherichia coli* among Children Aged between 0–36 Months in Peri-Urban Areas of Lusaka

**DOI:** 10.3390/microorganisms11112790

**Published:** 2023-11-17

**Authors:** Kapambwe Mwape, Samuel Bosomprah, Kennedy Chibesa, Suwilanji Silwamba, Charlie Chaluma Luchen, Nsofwa Sukwa, Cynthia Mubanga, Bernard Phiri, Mwelwa Chibuye, Fraser Liswaniso, Paul Somwe, Obvious Chilyabanyama, Caroline Cleopatra Chisenga, Monde Muyoyeta, Michelo Simuyandi, Tobias George Barnard, Roma Chilengi

**Affiliations:** 1Enteric Disease and Vaccine Research Unit, Center for Infectious Disease Research in Zambia, Lusaka P.O. Box 34681, Zambia; kapambwe.mwape@cidrz.org (K.M.); suwilanji.silwamba@cidrz.org (S.S.); nsofwa.sukwa@cidrz.org (N.S.); mwelwa.chibuye@cidrz.org (M.C.); chilengir@yahoo.com (R.C.); 2Water and Health Research Center, Faculty of Health Sciences, University of Johannesburg, P.O. Box 17011, Doornfontein 2028, South Africa; tgbarnard@uj.ac.za; 3Department of Basic Medical Sciences, Michael Chilufya Sata School of Medicine, Copperbelt University, Ndola P.O. Box 71191, Zambia; 4Department of Biostatistics, School of Public Health, University of Ghana, Accra P.O. Box LG13, Ghana; 5Next Generation Sequencing Unit and Division of Virology, Faculty of Health Sciences, University of the Free State, P.O. Box 339, Bloemfontein 9300, South Africa; 6Department of Biomedical Sciences, School of Health Sciences, University of Zambia, Lusaka P.O. Box 50110, Zambia; 7Amsterdam Institute of Infection and Immunity, Amsterdam University Medical Centers, 1105 AZ Amsterdam, The Netherlands; 8Department of Global Health, Amsterdam Institute for Global Health and Development (AIGHD), Amsterdam University Medical Centers, University of Amsterdam, 1105 AZ Amsterdam, The Netherlands; 9Division of Medical Microbiology, Department of Pathology, Stellenbosch University & National Health Laboratory Service, Tygerberg Hospital Francie van Zijl Drive, P.O. Box 241, Cape Town 8000, South Africa

**Keywords:** diarrhoea, children under five years old, diarrhoeagenic *E. coli*

## Abstract

Diarrhoea is a major contributor to childhood morbidity and mortality in developing countries, with diarrhoeagenic *Escherichia coli* being among the top aetiological agents. We sought to investigate the burden and describe the diarrhoeagenic *E. coli* pathotypes causing diarrhoea among children in peri-urban areas of Lusaka, Zambia. This was a facility-based surveillance study conducted over an 8-month period from 2020 to 2021. Stool samples were collected from children aged 0–3 years presenting with diarrhoea at five peri-urban health facilities in Lusaka. Stool samples were tested for diarrhoeagenic *E. coli* using the Novodiag bacterial GE+^®^ panel, a platform utilising real-time PCR and microarray technology to detect bacterial pathogens. Of the 590 samples tested, diarrhoeagenic *E. coli* were detected in 471 (76.1%). The top three pathogens were enteropathogenic *E. coli* 45.4% (*n* = 268), enteroaggregative *E. coli* 39.5% (*n* = 233), and enterotoxigenic *E. coli* 29.7% (*n* = 176). Our results revealed that 50.1% of the diarrhoeagenic *E. coli* positive samples comprised multiple pathotypes of varying virulence gene combinations. Our study demonstrates a high prevalence of diarrhoeagenic *E. coli* in childhood diarrhoea and the early exposure (<12 months) of children to enteric pathogens. This calls for the early implementation of preventive interventions for paediatric diarrhoea.

## 1. Introduction

Diarrhoea remains a major yet preventable public health threat among under-five-year-old children globally. In 2016, more than 400,000 deaths were attributed to diarrhoea among children below the age of five globally with low and middle income countries (LMICs) being the most affected due to poor sanitation, hygiene, and unsafe water supplies [1]. It is estimated that children between 0 and 36 months from LMICs suffer from an average of three diarrhoea episodes annually [2]. A review of various studies conducted in 34 sub-Saharan countries revealed an overall prevalence of 15.3% of diarrhoea among children under five between 2009 and 2018 [3].

There are several aetiological agents of diarrhoea ranging from viruses and bacteria to protozoa. Rotavirus and *Escherichia coli* are among the most common causative agents of diarrhoea among children in LMICs, including Zambia [4,5]. Although most *E. coli* exist as commensals in the gut microbiota, various pathogenic strains of the gut referred to as diarrhoeagenic *E. coli* (DEC) have been identified and are classified into pathotypes based on virulence traits [6]. The DEC pathotypes include enterotoxigenic *E. coli* (ETEC), enteropathogenic *E. coli* (EPEC), enterohaemorrhagic *E. coli* (EHEC), enteroaggregative *E. coli* (EAEC), and enteroinvasive *E. coli* (EIEC) [6]. These pathogens have been reported as major contributors to paediatric diarrhoea in several African countries [7,8,9,10]. Two major LMIC studies, the Global Enteric Multicenter Study (GEMS) and Malnutrition and Enteric Disease Study (MAL-ED) reported regional variations in the predominant DEC pathotypes and a significant proportion of these are of mixed aetiology [10,11].

In Zambia, hospitalisations due to acute gastroenteritis significantly dropped following the introduction of rotavirus vaccines from 40.9% in 2013 to 26.2% in 2014 [12]. Despite this drop in hospitalisations, diarrhoea remains a significant public health threat among children in Zambia and is one of the top five causes of morbidity and mortality among under-five-year-old children with an estimated prevalence of 15% [4]. In 2017 alone, 2900 deaths due to diarrhoea were recorded among children under five years of age [2]. Previous studies have placed *E. coli* among the top three bacterial aetiological agents of diarrhoea among children in Zambia [5,13]. Although these organisms are routinely isolated from diarrhoeal stool samples, the identification of the pathotypes is not made as it requires additional molecular assays that are not available due to limited resources. Due to the lack of routine DEC diagnosis, data on the burden posed by these pathogens among Zambian children are limited. The effective control of diarrhoea in any community requires an understanding of the circulating pathogens to allow for effective targeted therapy. Given the previous 2016 hospital-based report of circulating DEC strains with a prevalence of 18% in children with diarrhoea in Zambia [13], it is critical to gain an understanding of the burden posed by these pathogens.

Data on circulating enteric pathogens among children are critical to guide the implementation of preventive interventions. In this study, we report on the burden of DEC among children aged 0–36 months to determine their role in paediatric diarrhoea. We used the Novodiag bacterial gastroenteritis (GE+)^®^ panel on the Novodiag (Mobidiag, Espoo, Finland) [14], a platform that utilizes real-time PCR and microarray technology to identify pathogens [14].

## 2. Materials and Methods

### 2.1. Study Design and Participants

This was a facility-based surveillance study of children aged between 0–36 months presenting with diarrhoea at five health facilities located in peri-urban areas (i.e., Chainda, Chawama, Matero, Kanyama, and George) of the Lusaka district. A case of diarrhoea was defined as 3 or more episodes of looser than normal stools (softer or watery) in a period of 24 h. All the health facilities but one offer both in-patient and out-patient facilities and cater to a population of not less than 25,000 people. Sociodemographic data including age, sex, WASH, and household income were collected during the census where children were identified and enrolled. Clinical data were collected for each diarrhoea episode when the child reported to the facility. This included presenting symptoms (fever, stomach cramps, vomiting, and dehydration), duration and frequency of symptoms, anthropometrics, nutritional history, and case management and treatment. The participants were treated according to the standard treatment guidelines in Zambia. The Zambian standard treatment guidelines are based on the WHO’s integrated management of childhood illnesses (IMCI). The attending clinician determines the level of dehydration as ‘no dehydration’, ‘some dehydration’, and ‘severe dehydration’ based on clinical features. The child is then rehydrated using oral rehydration salts and/or intravenous fluids depending on the level of dehydration. In addition, zinc supplements are administered and breastfeeding is continued. Antibiotics are only prescribed under specific circumstances as stipulated in the treatment guidelines [15]. Clinicians documented the general condition of the participants and assessed the severity of diarrhoea using an in-house developed diarrhoea severity scoring tool [16]. One stool sample was collected from each child for the detection of DEC pathotypes. The study was conducted between November 2020 and July 2021. Any child presenting with diarrhoea, 36 months old or younger, and accompanied by a legally authorised representative willing to provide written informed consent were included.

Ethical approval was obtained from the University of Zambia Biomedical Research Ethics Committee (UNZABREC) (Ref. No. 1091. 2020) whilst permission to conduct the study was given by National Health Research Authority (NHRA). Written informed consent was obtained from parents or guardians on behalf of the participants. The parents or the guardians were assured about the confidentiality of information collected from them and keeping the identity of participants anonymous.

### 2.2. Study Procedures

*Stool Specimens:* Following informed consent, stool samples were collected in sterile sample collection containers and transported at 4–8 °C to the Microbiological laboratory for processing within 24 h of collection. The stool was transferred to an eNAT^®^ tube containing lysis buffer and preservative media (Copan Diagnostics, Murieta, CA, USA) using a Floq Swab (Copan Diagnostics, Murieta, CA, USA). The sample in the eNAT^®^ tube was then mixed by vortexing and stored at −80 °C for testing at a later date.

*Novodiag Bacterial GE+ testing:* The stool samples were processed according to the manufacturer’s instructions using the Novodiag bacterial GE+^®^ panel (Mobidiag, Espoo, Finland). This is a qualitative multiplex nucleic acid-based assay that uses real-time PCR and microarray technology to detect up to 14 bacterial pathogens in stool with a short turnaround time of less than two hours. The DEC pathotypes that are detected on this platform include: EPEC (*eae*), ETEC (*eltA*, *est*), EAEC (*aggR*), EIEC (*ipaH*), and EHEC (*eae*, *stx1* and *stx2*). Additionally, the Novodiag bacterial GE+ (Mobidiag, Espoo, Finland) panel detects *Campylobacter coli*, *Campylobacter jejuni*, *Clostridium difficile*, *Salmonella* spp., *Yersinia enterocolitica*, *Vibrio cholerae*, *Vibrio parahaemolyticus*, and *Yersinia pseudotuberculosis/pestis.* The Novodiag bacterial GE+^®^ (Mobidiag, Espoo, Finland) has been validated and shown to be a suitable platform for use in the routine rapid molecular diagnosis of gastroenteritis [14]. Briefly, before testing, the sample was allowed to thaw and then mixed by vortexing for 5 s. A volume of 600 µL of the sample was added to the bacterial GE+ cartridge (Mobidiag, Espoo, Finland). The cartridge was then capped tightly and loaded into the Novodiag^®^ instrument (Mobidiag, Espoo, Finland). The sample was run for 70 min after which qualitative results indicating the presence or absence of a pathogen were obtained and read.

#### Statistical Analyses

The sample size was based on estimating the prevalence of DEC with a certain level of precision. A single-group design was used to obtain a two-sided 95% confidence interval (CI) for a single proportion. The simple asymptotic formula was used to calculate the CI. The sample proportion of any of the DEC pathogens was assumed to be 0.5. To produce a CI with a width of no more than 0.1, 385 subjects were needed. The sample size was computed using PASS 2023, version 23.0.2.

Social demographics were summarised using frequencies and proportions to show the distribution across different subgroups. We estimated the prevalence of each pathogen and the associated 95% CI, as the proportion of participants with each pathogen out of the total number of children sampled. The association between social demographic characteristics and prevalence of each pathogen was tested using chi-square (or Fisher’s exact test where appropriate). Statistical significance was set at *p*-value < 0.05, and all statistical analyses were performed using Stata 14.2 (Stata Corp, College Station, TX, USA).

## 3. Results

### 3.1. Study Profile

A total of 1100 infants presented to the facility with diarrhoea. Of these, 590 infants had stool samples randomly selected for testing, thus ensuring that each stratum (age group, health facility, and month of collection) was accurately represented.

### 3.2. Risk Factors for DEC Infection

Males accounted for 55.4% of the infants with a larger proportion (42.9%) in our study being between 12–23 months of age. We also noted that 74.5% of our participants had access to a private toilet facility while about a third 29% had water piped into the house. In our study, 86.9% of the infants were breastfeeding. Of the 590 infant stool samples tested, 79.6% (*n* = 471) were positive for at least one DEC pathotype (Table 1). In our secondary analysis, we used a logistic regression to determine the effect of background characteristics on infection with DEC (Appendix A). We observed that children between 12–23 and 24–36 months of age were less likely to have EAEC infection with adjusted odds ratios (AOR) of 0.39 and 0.25, respectively. Children aged 12 months and above had an increased risk of ETEC and EIEC infection. Children whose mothers had attained post-secondary education were less likely to have EPEC infection than those with no formal education (AOR 0.15). Additionally, children in the highest social economic status group had a decreased risk of EPEC (AOR 0.59) and higher risk of ETEC (AOR 2.21) infections than those in the lowest social economic status group. There was generally no significant difference in the risk of DEC infections based on sex, caregiver, caregiver’s age, breastfeeding, household head’s marital status, source of drinking water, and type of toilet facility, as evidenced by the non-significant odds ratios (above 0.05) (Appendix A).

### 3.3. Pathogen Prevalence

We observed that EPEC was the most frequently detected pathotype 269 (45.3%) followed by EAEC at 39.4% and ETEC at 29.6%. Enteroinvasive *E. coli*/*Shigella* spp. ranked fourth with a prevalence of 20.1%. We observed that EHEC was detected in less than 10 samples with a prevalence of 1.2% (Figure 1).

### 3.4. Burden of Mixed Pathotypes

We noted that 233/592 (39.7%) of the participants tested positive for one pathotype in our study while the remainder of the participants had multiple DEC pathotypes detected. About 153/592 (25.8%) of the participants were positive for two pathotypes, 69/592 (11.7%) were positive for three pathotypes while 14/592 (2.4%) were positive for four pathotypes (Figure 2). In the same figure, the occurrence of each pathotype in combination with others was higher in proportion than as single pathotype. Furthermore, we noted that the EPEC–EAEC (51/153 (33.3%)) combination was the most prevalent in the samples with two pathotypes followed by EPEC–ETEC (37/153 (24.1%)) and then EPEC–EIEC/*Shigella* spp. (26/153 (16.9%)). In the three pathotype combination, EPEC–EAEC–ETEC (28/69 (40.5%)) was the most prevalent pattern. All pathotypes except EHEC were part of the four pathotype combination EPEC–EAEC–ETEC–EIEC/*Shigella* spp. (14/14 (100%)) (Appendix A). We recorded seven severe cases of diarrhoea among the children with mixed infections. Further analysis to determine whether there were additional factors associated with potential mixed infections revealed no significant associations (Appendix A).

## 4. Discussion

Despite being preventable, diarrhoea remains a significant scourge in children’s well-being and health in LMICs. Our study reports on the burden of DEC among infants presenting with diarrhoea in various outpatient facilities in Lusaka, Zambia. We noted that most of the diarrhoeal stool samples tested positive for at least one DEC pathotype, reaffirming the critical role that bacteria such as DEC play in paediatric diarrhoea infections in Zambia. This finding in particular highlights that diarrhoea aetiology goes beyond rotavirus, which has been the most implicated cause of diarrhoea in children. This aligns with recent research that has shown that bacteria account for a huge diarrhoeal burden among children in LMICs [10,11,17]. We also found that around 20% of the samples were negative for all the pathogens screened for and attributed these episodes to possibly other bacterial, viral, protozoal, or non-infectious aetiologies. Most of the infants in our study were reported to be breastfeeding; however, we observed an overall high prevalence of DEC among them. Although breast milk is known to have a protective effect against infectious diarrhoea among children, other factors such as suboptimal breastfeeding and poor hygienic practices among caregivers may contribute to diarrhoea among infants [17,18]. Although not significantly associated with DEC positivity, we noted that more than half of the children presenting with diarrhoea did not have water piped into the yard but rather used public taps or wells. This presents a problem of the higher faecal contamination of water, thereby exposing children to enteric pathogens, as reported in a previous study conducted in rural Zambia [19]. This study further reported that households with water piped into their yards reported less diarrhoeal episodes among children as compared to those without this facility [19].

Age-stratified prevalence data revealed that various pathotypes were higher in certain age groups, thus highlighting the infants most vulnerable to particular DEC pathogens. Enteroaggregative *E. coli* was higher among children under 12 months of age and was the second most prevalent pathogen being detected in more than 30% of the samples. This pathotype is reported to be responsible for both acute and persistent diarrhoea globally [20]. Two studies conducted in South America, in Brazil and Peru specifically, had similar findings with reports of EAEC being highly prevalent in this age group [21,22]. Unlike our findings, another study conducted in Egypt reported that children aged between 12–24 months were the most affected by EAEC [23]. Furthermore, these findings are unlike two previous reports from Lusaka, which both reported the prevalence of EAEC being less than 5% among children under 5 years of age [5,13]. The high exposure to DEC in young infants suggests that any interventions, i.e., vaccines aimed at preventing infection by this pathotype must be implemented quite early in the life of infants.

Furthermore, our study revealed a high prevalence of EIEC/*Shigella* spp., which is a known causative agent of bacillary dysentery. [24]. This finding is not uncommon as these pathogens are highly associated with diarrhoea among children below the age of five in LMICs [10,11,25]. One limitation in the detection of EIEC on the Novodiag bacterial GE+ is that the target is the *ipaH* gene, which is found in both *Shigella* spp. and EIEC. This limitation hinders the evaluation of the true burden of either pathogen; therefore, we could not conclusively attribute this prevalence to EIEC. Considering this limitation, our findings are in accordance with the previous findings of a study conducted among Zambian children presenting with moderate-to-severe diarrhoea that reported *Shigella* prevalence to be above 30% [5]. In the said study, similar to the platform used in the current work, the target gene for the detection of *Shigella* spp. is the *ipaH* gene.

Our study found that ETEC and EIEC prevalence was higher in the 12–23 months age group, and logistic regression revealed that children in this age group had a higher risk of being infected by the two pathogens than those below 12 months. Children in this age group are generally receiving mixed feeding and are thus exposed to more pathogens [26]. We also attributed the high prevalence of these pathogens in this age group to the possible waning of maternal antibodies that has been demonstrated to occur by this age in our population, hence reducing protection in this age group [27]. There was a general decline in the pathogen detection frequency in the older age group. This is likely due to repeated exposure, which induces acquired natural immunity providing more protection to these pathogens at this age. Children of middle–high socioeconomic status and those whose mothers had attained post-secondary education had a lower risk of EPEC infection. We attributed this finding to the fact that these two factors are likely to improve health and sanitation practices in households. Mothers who attained a higher education are more likely to have knowledge of transmission and implement preventive measures against enteric infections [28].

Our findings show that the most prevalent pathogens were EPEC, EAEC, and ETEC. Similar to previous reports from other African countries, including Kenya [18], Mozambique [29], Sudan [7], and Rwanda [26], about half the samples tested were positive for at least one DEC pathotype. Although these pathogens are not routinely screened for in Zambia due to the requirement of molecular tools for definitive diagnosis, recent research including our work affirms that DECs significantly contribute to diarrhoea among children. A previous hospital-based study by Chiyangi and colleagues in 2018 in Zambia reported a much lower prevalence of DEC among children, suggesting that these pathogens may not cause severe illness warranting hospitalisation but are a cause for concern among children in the community [13]. However, the higher prevalence of DEC in our study may also reflect changing trends in the predominant enteric pathogenic bacteria in our population. We reported EPEC as the most prevalent pathogen detected among the children in our study. Similar findings of EPEC being increasingly the predominant pathogen among children presenting with diarrhoea in various studies have been reported globally [30,31]. A limitation of note in the method used in our study is that the platform does not discriminate between heat-labile (LT) and heat-stable (ST) toxin-producing ETEC strains; therefore, additional characterization methods with individual toxin gene targets would have to be used to identify the toxin profiles of ETEC positive samples. We observed that EHEC was the least prevalent pathotype, and this finding is similar to a previous study conducted among children under five in Zambia that did not detect any EHEC [13]. The Novodiag bacterial GE+ detects all the standard target genes for EHEC detection; therefore, we consider our findings to be an accurate representation of the prevalence of this pathotype in this population [32,33].

The DEC pathotypes detected in our study occurred more in combination with 1 or more other pathotypes per sample than as single pathotypes. This finding agrees with several studies conducted in Africa, revealing a high pathogen burden in individual stool samples [34,35,36]. The occurrence of more than one pathotype, however, may be either a co-infection or single infections with hybrid DEC pathotypes [37]. This is a huge cause for concern as it affects the management of moderate to severe diarrhoea and may result in the inadequate treatment of diarrhoea cases. Furthermore, the high prevalence of multiple pathotypes per sample reported in our study requires further research to definitively define the causative agent of a diarrhoeal episode in our setting. This is crucial for driving preventive measures that are to be put in place, such as the identification of enteric vaccine intervention needs of the children in our setting. Studies have shown that beyond dehydration during diarrhoeal episodes, high exposure to enteric pathogens such as DEC among infants is of great concern as these pathogens are reportedly associated with poor growth trajectory in younger children [38,39,40,41]. 

## 5. Conclusions

This study highlights the prevalence and early exposure to DEC among children in Zambia. Enteropathogenic *E. coli*, EAEC, and ETEC are the most prevalent pathotypes affecting children presenting with diarrhoea in Lusaka, Zambia. The infections are poly-microbial in nature and warrant further investigations in terms of how this impacts environmental enteric dysfunction (EED), children’s growth velocity, oral vaccine uptake, and other sequelae associated with early and repeated exposure to these pathogens. Future investigations should incorporate both outpatients and inpatients to determine the clinical significance of the pathotypes in association with diarrhoea severity. Additionally, molecular tools for the detailed characterisation of DEC (antimicrobial resistance and virulence genes) must be utilized in future investigations.

## Figures and Tables

**Figure 1 microorganisms-11-02790-f001:**
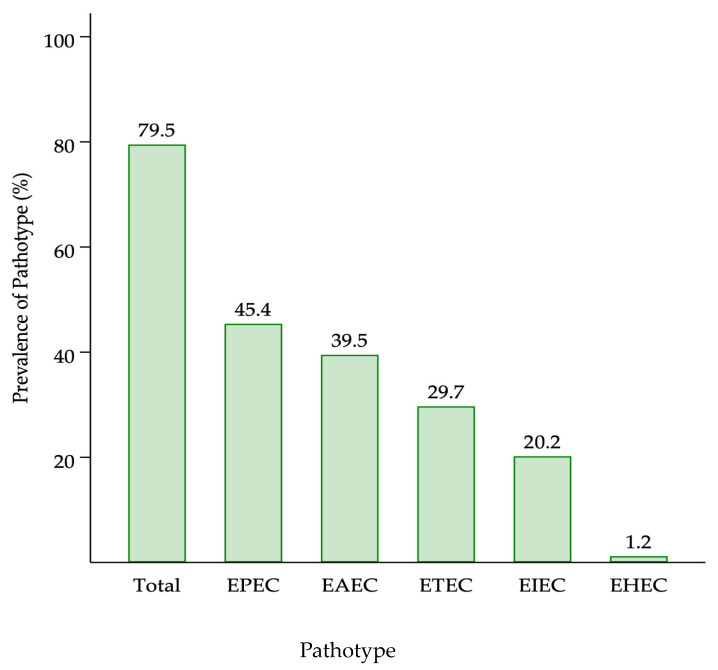
Prevalence of DEC pathotypes detected. EPEC—enteropathogenic *E. coli*; EAEC—enteroaggregative *E. coli*; ETEC—enterotoxigenic *E. coli*; EIEC—enteroinvasive *E. coli*; EHEC—enterohaemorrhagic *E. coli*.

**Figure 2 microorganisms-11-02790-f002:**
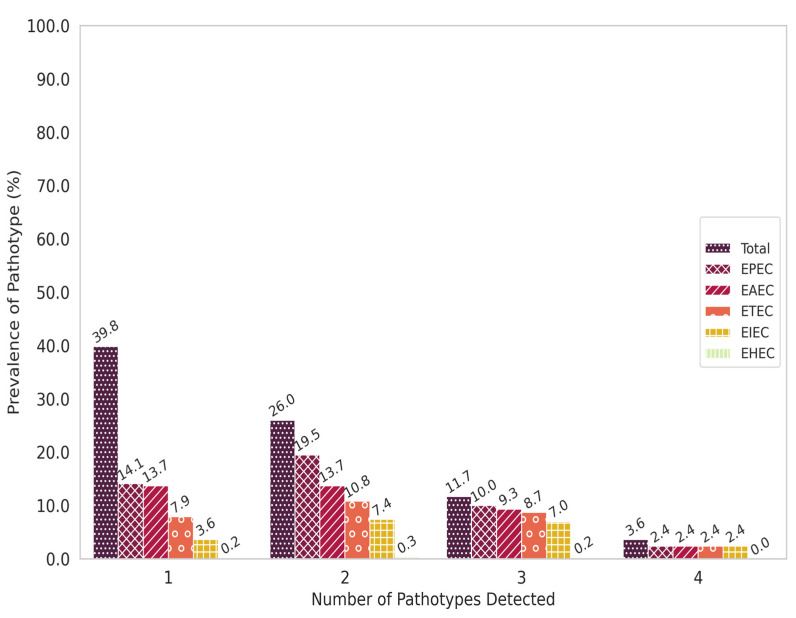
Prevalence by number of pathotypes detected per sample. The figure illustrates the proportion of each pathotype in the different infection patterns based on the number of pathotypes detected in each sample. The blue bars represent the prevalence of all the pathotypes consolidated while the other bars each represent a single pathotype. The *Y*-axis shows the prevalence of the pathotypes while the *X*-axis shows the number of pathotypes detected in each sample.

**Table 1 microorganisms-11-02790-t001:** Demographic and clinical details of study participants.

Characteristic		Total *N* = 590	DEC Positive (*n* = 471, 79.6%)				*p*-Value	EPEC Positive (*n* = 268, 45.4%)	*p*-Value	EAEC Positive (*n* = 233, 39.5%)	*p*-Value	ETEC Positive (*n* = 175, 29.7%)	*p*-Value	EIEC/*Shigella* spp. Positive (*n* = 119, 20.2%)	*p*-Value	EHEC Positive (*n* = 7, 1.2%)	*p*-Value
		***n* (% of total)**	***n* (%)**					***n* (%)**		***n* (%)**		***n* (%)**		***n* (%)**		***n* (%)**	
Age (Months)	<12	229 (38.8)		183 (79.9)			0.586	99 (43.2)	0.496	126 (55.0)	<0.001	49 (21.4)	0.002	25 (10.9)	<0.001	1 (0.4)	0.317 *
	12–23	253 (42.9)		204 (80.6)				122(48.2)		82 (32.4)		91 (36.0)		60 (23.7)		5 (2.0)	
	24–36	108 (18.3)		82 (75.9)				47 (43.5)		25 (23.1)		35 (32.4)		34 (31.5)		1 (0.9)	
Sex	Male	327 (55.4)		260 (79.5)			0.990	152 (46.5)	0.564	124 (37.9)	0.384	99 (30.3)	0.716	60 (18.3)	0.219	5 (1.5)	0.324*
	Female	263 (44.6)		209 (79.5)				116 (44.1)		109 (41.4)		76 (28.9)		59 (22.4)		2 (0.8)	
Caregiver	Mother	505 (85.6)		400 (79.2)			0.678	232 (45.9)	0.539	199 (39.4)	0.917	143 (28.3)	0.081	101 (20.0)	0.803	6 (1.2)	0.733 *
	Other	85 (14.4)		69 (81.2)				36 (42.4)		34 (40.0)		32 (37.6)		18 (21.2)		1 (1.2)	
Breast Feeding	No	28 (4.7)		22 (78.6)			0.983	13 (46.4)	0.970	14 (50.0)	0.204	8 (28.6)	0.985	5 (17.9)	0.460 *	0 (0.0)	0.724 *
	Yes	508 (86.1)		400 (78.7)				234(46.1)		193 (38.0)		146(28.7)		106(20.9)		6 (1.2)	
	Missing	54 (9.2)		47 (87.0)				21 (38.9)		26 (48.1)		21 (38.9)		8 (14.8)		1 (1.9)	
Caregiver’s age	18–24	148 (25.1)		115 (77.7)			0.883	69 (46.6)	0.673	55 (37.2)	0.808	41 (27.7)	0.703	30 (20.3)	0.494	0 (0.0)	0.251*
	25–34	232 (39.3)		182 (78.4)				106(45.7)		94 (40.5)		72 (31.0)		49 (21.1)		4 (1.7)	
	35+	74 (12.5)		56 (75.7)				30 (40.5)		29 (39.2)		20 (27.0)		11 (14.9)		1 (1.4)	
	Missing	136 (23.1)		116 (85.3)				63 (46.3)		55 (40.4)		42 (30.9)		29 (21.3)		2 (1.5)	
Household Head’s Marital Status	Married	474 (80.3)		372 (78.5)			0.466	222 (46.8)	0.153	181 (38.2)	0.418	137 (28.9)	0.321 *	99 (20.9)	0.044	4 (0.8)	0.095 *
	Single	23 (3.9)		19 (82.6)				12 (52.2)		10 (43.5)		5 (21.7)		8 (34.8)		1 (4.3)	
	Divorced/ Separated/Widowed	93 (15.8)		78 (83.9)				34 (36.6)		42 (45.2)		33 (35.5)		12 (12.9)		2 (2.2)	
Mother’s Highest Level of Education	No formal education	18 (3.1)		16 (88.9)			0.669 *	13 (72.2)	0.017 *	6 (33.3)	0.799	7 (38.9)		3 (16.7)	0.859 *	0 (0.0)	1.000 *
	Primary	164 (27.8)		129 (78.7)				82 (50.0)		63 (38.4)		47 (28.7)		31 (18.9)		2 (1.2)	
	Secondary/Post-Secondary	408 (69.2)		324 (79.4)				173(42.4)		164 (40.2)		121 (29.7)		85 (20.8)		5 (1.2)	
Social Economic Status	Lowest	226 (38.3)		176 (77.9)		0.886		116(51.3)	0.043	89 (39.4)	0.969	6 (24.8)	0.087	38 (16.8)	0.180	4 (1.8)	0.513*
	Middle	206 (34.9)		164 (79.6)				92 (44.7)		81 (39.3)		60 (29.1)		49 (23.8)		1 (0.5)	
	Highest	113 (19.2)		90 (79.6)				42 (37.2)		43 (38.1)		41 (36.3)		25 (22.1)		1 (0.9)	
	Missing	45 (7.6)		39 (86.7)				18 (40.0)		20 (44.4)		18 (40.0)		7 (15.6)		1 (2.2)	
Source of Drinking Water	Piped into house/ yard/well	172 (29.2)		129 (75.0)		0.296		72 (41.9)	0.365	66 (38.4)	0.906	42 (24.4)	0.164 *	43 (25.0)	0.094 *	1 (0.6)	0.187 *
	Public tap/ borehole	355 (60.2)		287 (80.8)				171(48.2)		141 (39.7)		111(31.3)		67 (18.9)		4 (1.1)	
	Other	17 (2.9)		13 (76.5)				7 (41.2)		6 (35.3)		3 (17.6)		1 (5.9)		1 (5.9)	
	Missing	46 (7.8)		40 (87.0)				18 (39.1)		20 (43.5)		19 (41.3)		8 (17.4)		1 (2.2)	
Toilet Facility	Shared	99 (16.8)		79 (79.8)		0.761		46 (46.5)	0.823	40 (40.4)	0.809	32 (32.3)	0.336	18 (18.2)	0.576	1 (1.0)	0.696 *
	Private	440 (74.6)		345 (78.4)				199 (45.2)		172 (39.1)		121 (27.5)		91 (20.7)		5 (1.1)	
	Missing	51 (8.6)		45 (88.2)				23 (45.1)		21 (41.2)		22 (43.1)		10 (19.6)		1 (2.0)	

* Fisher’s exact test.

## Data Availability

Data will be made available upon reasonable request to the corresponding author.

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
