# Peer review of "Prevalence of Diarrhoeagenic Escherichia coli among Children Aged between 0–36 Months in Peri-Urban Areas of Lusaka"

_microorganisms, 2023, doi:10.3390/microorganisms11112790_

Round 1

Reviewer 1 Report

Comments and Suggestions for Authors

The authors conducted a study to determine the prevalence of diarrhoea-inducing E. coli in Lusaka, Zambia, with a particular focus on children under 3 years old. Using stool samples, they applied a PCR-based technique known as Novodiag to detect various virulence genes. The presence of these genes characterizes pathogenic E. coli.

Their primary discovery was that over 70% of the samples contained diarrhoea-causing E. coli. The most common E. coli pathotypes identified were EPEC, EAEC, and ETEC. Understanding the prevalence of different E. coli pathotypes worldwide is crucial. It guides where interventions, including vaccine development against these E. coli strains, are most urgently needed and underscores the importance of ongoing surveillance studies.

I have comments that I think can improve the manuscript:

Line 39: The method that was used cannot say that these are co-infections. See comment about this further down.

Line 70: Change 2900 to 2,900

Line 71: … recorded among under five children” à “…. recorded among children under five years of age”

Line 83: Not a complete sentence. Please correct.

Line 92: I think it would be better if you move line 97 here as I was wondering what the definition of loose stool was.

Line 95-96: I think it would be very helpful for other researchers in the field to understand what the standard treatment guideline in Zambia for diarrhoea is.

Line 120: Does Novodiag differentiate between STh and STp? Indicate if it does or not.

Figure 1. I don’t think the figure is needed.

Table 1: Do you have any information about the severity of the diarrhoea? Is this something you can look at? Also, what does * indicate? Please add this to a footnote.

Line 159, line 161, line 168: Remove extra spaces.

Figure 2: Why are there error bars? You can add the precise % of each of the pathotypes in the figure. You could also add a bar to show the total. I’d expand the figure text as you are suppose to understand the figure without reading the text.

Line 181: Any factors identified to be associated with potential mixed-infections?

Section 3.5. I had a comment in the start about whether Novodiag can detect whether a sample contains multiple pathotypes or what it really does is to look at the presence of genes used to characterise pathotypes. So, it can be mixed-infections but it could also be hybrid E. coli pathotypes. I think it is important to mention this. To say anything about mixed infections you need downstream analysis.

Figure 3a: It would be more interesting to show not only the frequency of samples that are positive for 1, 2, 3 or 4 pathotypes but to know what combinations you found. I would not use “Number of infections” as a y-axis title. It’s number of samples positive for 1-4 pathotypes.

Figure 3b: The footnote on P-value needs to be bigger. You could make a plot that show the prevalence of potential co-infections but also show that the different combinations. In summary, I think you can make a better figure summarising the results shown in 3a and 3b.

Comment: Do you know the prevalence of LT and ST for ETEC positive samples? I think this is of importance.

Line 206: Not a quarter of the samples were negative, but 20.2%.

Line 217-218: What does “into the yard” mean. Are there multiple households using the same tap or is it one household? Is the water cleaner (contains less fecal contamination) from piped taps into the yard compared to public taps?

Line 225: Add a space between globally and ref 19

Line 229: Clarify, is it 5% in total or 5% in children <12 months of age?

Line 233: Replace Of note also à Furthermore, notable or interestingly

Line 243: ipaH should be in italics.

Line 245: change to this: “… revealed that children in this age group ….”

Line 248: should it be weaning?

Line 253-254: I would not use the word “protective” here.

Line 264: When was the study by Chiyangi and colleagues conducted – it is important as the authors state that prevalence in this study is different and potentially due to a shift in the most prevalent DECs.

Line 273-274: Not convinced that all samples positive for multiple pathotypes are co-infections but could be hybrid pathotypes. I’d formulate it slightly differently.

Line 289: Declare the abbreviation EED.

Reviewer 2 Report

Comments and Suggestions for Authors

The manuscript submitted by Mwape et al. presents results on the search for DEC in children's stool samples. The central finding is related to the amount of ETEC detected. The manuscript is well written, as is the presentation of the results. Some comments are attached to the document.

Although it is at the end of the manuscript, I would like to inform you of the individual participation of the 17 authors of the manuscript.

Reviewer 3 Report

Comments and Suggestions for Authors

Line 39& 243: Italicize E. coli & ipaH gene

Line 83: “We the Novodiag bacterial gastroenteritis (Ge+) panel”: should be “We used the Novodiag bacterial gastroenteritis (Ge+) panel “

Lines 117-118 “It is a qualitative multiplex nucleic acid-based assay that uses real-time PCR and microarray technology to detect up to 14 bacterial pathogens in stool”: What are these pathogens?

Line 193 “while the X-axis shows the of infecting” delete “of”

Lines 182-186: Authors should write the identified pathogens in the mixed infections

Line 289 “EED” write in full at the first mention.

All species should be italicized in the references

Comments on the Quality of English Language

Minor editing is required

Round 2

Reviewer 1 Report

Comments and Suggestions for Authors

Comments for authors

Thank you for the updated version. I do have a few additional comments.

Line 54: Spell out 3 -> three.

Line 81-82: You write about a hospital based report, but don’t mention what it showed or am I missing something here?

Line 98: don’t write etc

Line 103-109: Out of curiosity, if needed is antibiotics administered and are there guidelines to which antibiotic that should be used?

Line 133:lt and st are not the gene names, if you want the gene names it should be LT = eltAB (not sure about the novodiag is looking at eltA and/or eltB. For ST (not discriminating between STh, STp or STb) you can use est.

Figure 1: You can make the figure smaller with narrower bars. Also, in the legend you need to write E. coli in italics.

Line 205: I think that there is an unclosed bracket here (Figure 2. In ….

Line 206-211: I am not sure what these numbers are. You need to clarify that. For EPEC-EAEC, 51 out of how many? Also, I would suggest you write EPEC/EAEC instead. Lastly, you need to write the numbers in a consistent manner – compare Line 207 with line 211.

Figure 2: You need to adjust the figure to align with colorblindness. The numbers on top of each bar is a bit cramped. You can write them at a 45 degree angle instead. I’m not sure about this figure. Would it be more informative to actually just plot the different pathotype combinations?

Line 222: scourge?

Line 255: i.e -> i.e.,

Line 257: Need to rephrase the sentence, e.g. Furthermore, the high….

Line 263: Should you throughout the text potentially write EIEC/Shigella spp. instead?

Line 295: Lt and St -> I’m not sure if you are talking about the genes eltAB/est or the proteins/toxins LT and ST.

Line 301-302: Check spelling and rephrase the sentence.

Line 307: Here you write co-infection. Instead write the high prevalence of multiple pathotypes detected in individuals samples or something… You obviously mention that these could be co-infections.

Comments on the Quality of English Language

All my comments can be found in the attached file.

Reviewer 2 Report

Comments and Suggestions for Authors

In this new version of the manuscript, the authors have made all the suggested changes. Therefore, I suggest approving this manuscript for publication in Microorganisms.

Author Response

Thank you for taking the time to review the manuscript. Your feedback was very helpful.